# Two-Leak Case Diagnosis Based on Static Flow Model for Liquid Transmission Pipelines

**DOI:** 10.3390/s23187751

**Published:** 2023-09-08

**Authors:** Pawel Ostapkowicz, Andrzej Bratek

**Affiliations:** 1Faculty of Mechanical Engineering, Bialystok University of Technology, Wiejska 45C, 15-351 Bialystok, Poland; 2ŁUKASIEWICZ Research Network-Industrial Research Institute for Automation and Measurements PIAP, Al. Jerozolimskie 202, 02-486 Warsaw, Poland

**Keywords:** pipelines, leak detection, multiple leakages, static flow model

## Abstract

The article deals with a diagnosis of multiple leaks from liquid transmission pipelines using analytical methods. Such solutions, based on advanced mathematical models of pipeline flow dynamics, usually turn out to be very complex and time-consuming. However, under certain operating conditions, a simpler approach may also be useful. Such an idea is presented in this paper, proposing two simplified methods for diagnosing double leakages. In principle, these methods apply to both simultaneous and non-simultaneous leaks. The first one uses a static model of a pipeline involving two leaks and takes advantage of the minimization of the objective function defined as the squared deviation of the modeled pressures from the pressures measured on the pipeline. The second method uses a pipeline flow model of a static type in combination with a gradient indicator aimed at detecting leaks and employing algorithms assigned to determining the location and size of leaks. The results of methods’ validation, based on tests carried out with the use of measurement data obtained from an experimental water pipeline, were also presented. The outcomes of the performed tests proved the methods’ effectiveness in terms of detection, isolation, localization, and intensity estimation of both simultaneous and non-simultaneous double leakages.

## 1. Introduction

The basic requirement for the safe operation of transmission pipelines is to maintain tightness, which specifically applies to the main section of the pipeline. On the one hand, the above requires preventive measures that include monitoring the technical condition of the pipeline. On the other hand, it means that the pipeline must be equipped with an appropriate leak detection and identification system (LDIS).

Commonly used solutions of LDIS are the ones applying the so-called analytical (internal, indirect, or software-based) methods. These methods involve measurements of flow parameters in the pipe, such as flow rate, pressure, and temperature [1,2,3,4]. In addition, it can be mentioned here that the leak monitoring systems often utilize another category of diagnostic methods, which also give possibilities for detection and localization of single as well as multiple leakages, but without clear identification of their intensities as in the case of the above-mentioned analytical methods. Generally, these so-called external, direct, or hardware-based methods are characterized by leak disclosing from outside of the pipe, using for this purpose special devices, including microphones, hydrocarbon detectors, thermal cameras [2,3,4]. Such methods are not considered in this paper.

LDIS should meet the expectations and requirements of pipeline operators by detecting, localizing, and estimating the size of multiple leakages, in addition to detecting single leakages. The most common case of multiple leakages are double leakages.

Double leakages, owing to their timing (occurrence), are classified into concurrent or non-concurrent. Internal methods for diagnosing multiple (double) leaks are reviewed in several papers [1,5,6,7], with references to both non-simultaneous [8,9] and simultaneous leak cases [10,11,12,13].

On the whole, software-based detection and localization of multiple leaks, even in the two-leak case, remains a challenge [9,14]. As regards the difficulty, non-simultaneous leaks are easier to diagnose than simultaneous or almost simultaneous leaks in which transient states caused by their occurrence overlap in time. In practice, in the case of non-simultaneous leaks, known methods, such as wave pressure [15] typically used to detect and localize single leaks, can sometimes be applied. This is only possible when the leaks are delayed enough that they can be treated separately as single leaks. However, in case of simultaneous leaks, the straightforward application of diagnostic methods, which were conceived in principle for single leak incidents, turns out to be inefficient. In particular, the approach may lead to incorrect localization results for both kinds of multiple leaks. Similarly, the application of gradient methods [16] may result in poor leakage positioning.

Therefore, more advanced solutions are applied, which often use the models of flow dynamics [1,8,9,10,11,12,13,14,17]. When a two-leak case is perceived from the perspective of dynamic models in steady-state conditions, its general symptoms are consistent with a case of a single leak [1]. Hence, after possible detection that the pipeline is leaky, identification (isolation) of two simultaneous leaks is not possible without analyzing changes in flow dynamics caused by their occurrence [12,14].

In practice, the use of analytical methods based on mathematical models of flow dynamics involves resolving numerous problems. To a large degree, they relate to aspects concerning the elaboration and finding solutions for the model. In order to ensure a high accuracy of estimation of the leak location, a dynamic flow model corresponding to a typical section of a transmission pipeline of 30–100 km in length should be composed of several dozen sections, as stated in [17]. This leads to an extensive system of equations that are hard to solve. Regardless of a freely numbered set of equations, the computation of the model’s outputs requires the introduction of adequately well-matched boundary variables, together with their reliable values. The boundary variables are usually the flow rates and pressures at the ends of the duct, but they may not be enough. Therefore, additional boundary variables, which are attributed to other sections of the dynamic model, are often needed. When determining the boundary conditions, the quality of data acquired from measuring devices mounted on a pipeline is crucial. In practice, the data samples are noisy and error-prone due to the complex and extensive structure of the measuring systems. Another significant problem arises from the insufficient dynamics of measurement systems in tracking real changes and values of measured process variables. Additionally, disturbances in the flow can falsify measurement data. As a result, incorrect measurement data provided in a model as boundary variables or those used for the calculation of reference variables do not make it possible to obtain a correct final diagnosis.

These issues are rather difficult to solve, which is noticeable for the implementation of various mathematical model-based methods developed to diagnose multiple leakages. For example, this concerns inverse analysis methods [18,19], the operating principle of which is to use a test signal to generate transient flow. By comparing model-generated data with measured data and accurately fitting the model (i.e., minimizing the obtained deviations), it enables the detection and location of both leaks. Similarly, these aspects are also important for so-called automatic control approach methods [1,8,9,10,12,17,20], which generally use the mathematical model of liquid flow dynamics described in state-space along with its further analysis based on the implementation of state observers.

As a matter of fact, the model-based methods can be more effective due to many solutions that are proposed in the literature in order to improve the usefulness of the mathematical flow model. Some of them are more widely dealt with in the second section of this work. Generally, such solutions focus on many aspects, such as the development of improved models using a more accurate solution to one-dimensional flow modeling [21], the division of a pipeline into a minimum number of sections with considered leak locations [8], the improved selection of the optimal discretization grid in space and time [22], appropriate selection of the boundary variables [23]. It can also be mentioned that estimation techniques are used for a number of process parameters, such as the coefficient of friction [1,24].

However, in the case of the previously mentioned inverse analysis methods, their implementation must meet many regimes and details and is associated with many uncertainties. It relates especially to diagnostic tests, which must be performed in order to generate a transient state in the flow, usually using fittings located at the extreme pipe section. For example, a transient flow can be generated by closing the downstream end valve, followed by an analysis of the propagation of the pressure wave through a pipe [19]. Similarly, a sudden closure of the shut-off valve located at the pipeline’s outlet can be used for the localization of a single [25] or two leaks [26] by considering the disclosure of partial reflections of pressure waves off the leak spots. Another test involves a specific valve operation that excites the fluid required for an approach based on frequency response analysis [18]. The input of a sine-like pressure signal generated by a pump unit after detecting leakage occurrence is used to perform the localization of two simultaneous leaks based on the analysis of the output signal obtained at the other end of a pipeline [13]. Generally, such solutions are not welcomed by pipeline operators due to potential danger caused by uncontrolled pressure changes in a pipeline and distortion of desirable stable flow conditions.

It cannot be overlooked that the implementation of analytical methods with a utilized dynamic flow model is rather complicated and time-consuming due to the relatively large amount of time required to solve the equations used in its composition. Additionally, such advanced methods involve high expenses resulting from the use of required equipment, including an appropriate number of very accurate sensors, data transmission systems with highly rigorous time synchronization, and powerful computers. Significant costs are also generated by their maintenance and service.

Accounting for the problems pointed above, the authors set to elaborate simplified solutions aimed at diagnosing double leakages disregarding the use of models dealing with pipeline flow dynamics.

Previously, the authors applied simplified methods to diagnose single leakages which occurred during the stable flow process in a pipeline. The methods proved effective even with significantly small leaks, the intensity of which varied from 0.1% to 1.0% of the nominal flow rate [27]. Furthermore, the proposed solutions for such type methods were successfully adapted to diagnose single leakages occurring during a transient state. This unsteady flow process resulted from a change in the operating point of a pipeline, which involved increasing the rotation velocity of a pump [28].

In the currently discussed topic, as a result of the author’s work, there are solutions presented in a shorter post-conference paper [29]. In the present article, we offer a more detailed discussion about these two proposed simplified methods, which are aimed at diagnosing two simultaneous and non-simultaneous leakages, including all diagnostic tasks, i.e., detection, localization, and size (intensity) estimation. The operational areas of the proposed methods, as regards the matter of two-leak events, concern in the case of the former ones, their occurrence in steady state conditions, whereas in the case of the latter ones, the occurrence of the first leakage is also considered in the same steady state, but the second leakage is assumed in a transient state resulting from the first leak occurrence. In general, the proposed methods use a combination of elaborated static flow models with appropriate techniques and algorithms for diagnosis and signal processing.

This work’s contribution consists in the presentation of both proposed methods and the results of their verification, including graphs that illustrate the exemplary courses of the values of the calculated locations and the size of leaks. These illustrated parameters show the real efficiency of a given method in time counting. Moreover, efficiency evaluation and discussion of the results are provided. Such assessment concerns the use of measurement data acquired from the experimental pipeline, examining the scope of the operational area of both proposed methods in reference to leak intensity of about 1% of the nominal flow rate.

This paper is organized as follows: The Section 2 initially presents a description of typically used dynamic and static flow models, including basic aspects related to their elaboration and application. The subsequent part presents both proposed methods aimed at double leak diagnosis. Next, we show the laboratory water pipeline and experiment program with simulated two concurrent and non-concurrent leaks. The Section 3 presents aspects of the practical implementation of both methods as well as the effects concerning detection, localization, and size estimation of simulated leakages, including the discussion of the obtained results. Additionally, we point out possibilities for further performance improvement of both proposed methods. Finally, conclusions are given in the Section 4.

## 2. Materials and Methods

Generally, analytical methods used for leakage diagnosis can be divided into not-model-based and model-based approaches. In the case of the latter ones, these methods can be based on the use of different solutions of mathematical models of flow process.

### 2.1. Dynamic and Static Flow Models Used for Transmission Pipeline

When modeling the flow process in the transmission pipelines for the purpose of diagnosing leakages, both dynamic and static mathematical flow models are utilized.

#### 2.1.1. Typical Solutions of Models

In general, there are two solutions used for each of such models. The former ones are designed without consideration of any leaks, whereas the latter ones include this issue.

Elementary aspects, which concern both dynamic and static flow models, are discussed using the example of a pipeline given below.

#### 2.1.2. Considered Configuration of Model Pipeline

A typical transmission pipeline installation used for pumping a liquid medium consists of a pipe, a pump system, initial and final tanks. However, let us consider only its main part, i.e., a pipe duct.

Let us consider a pipeline characterized by horizontal and straight-line laying, constant internal diameter of the pipe d(z)=const along its entire length L defined according to the coordinate z, and linearly elastic and lightly deformable walls of the pipe.

At the beginning, let us assume that this pipe duct is tight, the pumped fluid is slightly compressible, and the flow is one-phased and isothermal, as well as is characterized by constant value of the density ρ(z)=const. Next, let us consider the same pipeline with double leaks.

#### 2.1.3. Dynamic Flow Models without and with Double Leaks

To describe unsteady flow in the above-defined pipe duct equations based on the principles of conservation of mass (1) and momentum (2) are used.
(1)$$\frac{A}{\rho }\frac{\partial p(z,t)}{\partial t}+c^{2}\frac{\partial q(z,t)}{\partial z}=0$$
(2)$$\frac{\partial q(z,t)}{\partial t}+\frac{A}{\rho }\frac{\partial p(z,t)}{\partial z}+\mu q(z,t)\left|q(z,t)\right|=0$$
where p—pressure [Pa], q—flow [m^3^/s], z—pipeline’s length coordinate [m], t—time coordinate [s], ρ—liquid density [kg/m^3^], A—cross-sectional area of pipe [m^2^], c—pressure wave speed in fluid [m/s], and $\mu =\frac{\lambda }{2dA}$, where d—inner diameter of pipe, λ—friction factor.

By discretizing these two partial differential equations, they can be integrated into a distributed-parameter model. Different techniques are proposed in the literature for this purpose, which are also aimed at solving the model. Some examples include the centred difference scheme [1], the method of characteristic line [24], the finite difference method [20,23], the orthogonal collocation method [30]. Depending on the use of one of such methods, a dynamic model about a specific finite difference scheme can be obtained. In the case of any model, it is necessary to obtain the numerical stability of the calculations. This requires applying an appropriate discretization grid.

Generally, the discretization grid relates to the space and the time. In the space, it consists in the division of the pipeline of the total length L into n sections, which are defined by the grid variables Δz. Particular sections are separated by the nodes, which are crucial to loading the inputs and reading the outputs considered during the calculations. It is expected that the positions of some model’s nodes should be compatible with the locations of measuring sensors because it allows comparing real and computational data. Even though it is theoretically possible trying to make the division of the pipeline into sections with identical or different lengths, in practice, such expectations can turn out difficult to meet. It results from the conception adopted in order to describe the flow through the pipeline by means of Equations (1) and (2), which assume the phenomenon of acoustic (pressure) wave propagation as the main mechanism of all transient events (unsteady flows). A proper implementation of this principle means the necessity of obtaining specific relations between the discretization in space and the discretization in time that is defined by the step variables Δt. For example, such problems relate to the design of the dynamic flow models according to the widely used method of characteristics [1,19,24]. Here, it is required to fulfill the relation Δz≥cΔt between the space grid Δz and time step Δt defined as Courant–Friedrichs–Lewy condition, where c represents speed of the pressure wave. Therefore, determination of the correct values for both discretization steps Δz and Δt is especially important. In practice. such selection is usually performed in steps by trial and error, which poses a serious problem. In order to choose optimal discretization, the formula proposed in [22] can be useful since it takes into account the impact of physical parameters (geometry and physics) of the flow process and leads to the calculation of the value of the so-called Courant number a, a coefficient from the range (0, 1〉 defining the qualitative link between the two discretization steps, as follows $\Delta t=a\frac{\Delta z}{c}$.

As previously mentioned, dynamic flow models can include the modeling of leakages. When two leaks are present, they are modeled in the nodes using the orifice Equation (3).
(3)$$q_{u(1,2)}=f_{\left(1,2\right)}\sqrt{2g\frac{\left(p_{\left(1,2\right)}-p_{o}\right)}{\gamma }}$$
where $q_{u(1,2)}$—intensity (volumetric flow) of the first and the second leak, $f_{\left(1,2\right)}$—constant, respectively, for the first and the second leak, that depends on discharge coefficient and area of the orifice, $p_{\left(1,2\right)}$—pressure at individual leak point (inside the pipe), $p_{o}$—pressure (outside the pipe), γ—specific weight of a fluid.

When designing a dynamic model with two leaks, the rules for selecting space discretization given in [11] can be helpful. In a two-leak case, the minimal finite dimension of the dynamic model can consist of only three sections, often non-uniform.

Both types of dynamic flow models, with and without modeled leaks, are expected to reproduce the behavior of the transported fluid, especially in its dynamic aspect. In practice, this information is represented by the model’s outputs: flow rate and pressure calculated for selected sections and nodes.

For this purpose, any dynamic flow model with specific grid discretization that fulfils numerical stability requirements also needs correctly introduced parameters related to pipeline geometry, pumped fluid, and flow process. To achieve satisfactory reconstruction of flow dynamics, particularly pressure wave propagation, calculations should be made with a properly short time resolution, i.e., computational step defined as the interval between consecutive calculated values of model outputs.

However, to calculate outputs for any dynamic flow model, it is necessary to define boundary conditions and provide input values, including nominal flow rate and position and intensity of leaks.

As boundary conditions, known values of pressures or flow rates at pipe ends or a combination of both are usually used. All such configurations of boundary conditions are reviewed in the literature [23], also considering their selection to obtain the best model structure representing a pipeline system.

In the case of a dynamic model with added leaks, outflow associated with each leak produces discontinuity. Therefore, additional boundary conditions are introduced. Assuming that the intensities of both leaks are represented by the variables $q_{u1}$ and $q_{u2}$, then, flows in sections before (A) and after (B), their positions must be assumed as follows $q_{A(1,2)}=q_{u(1,2)}+q_{B(1,2)}$.

#### 2.1.4. Static Flow Models without and with Double Leaks

Static flow models are generally designed to model the flow in transmission pipelines operating under steady-state conditions. This is characterized by pressure and flow rate signals that remain unchanged over time or exhibit small amplitude variations around a defined static value.

Using this assumption, it is possible to derive a static flow model from partial differential Equations (1) and (2) by considering $\frac{\partial p}{\partial t}\to 0$ and $\frac{\partial q}{\partial t}\to 0$. This approach is presented in [31]. Analytical solutions are proposed for two cases of pipe inclination angle: zero or nonzero. These solutions allow for the computation of pressure distribution along a pipeline when the pressures at both ends and the inclination angle are introduced as boundary conditions.

However, there are also more conventional ways to develop a static flow model. One such method is to use Bernoulli’s equation, which describes a momentum-based force relation in frictionless flow between pressure, velocity, and elevation.

For a horizontally and straight-line arranged pipeline without leaks, with an incompressible steady flow and no line package effect, it can be assumed that the flow rates at both ends are identical, i.e., $q=q_{1(in)}=q_{2(out)}$. In this case, Bernoulli’s equation can be written in the form of Equation (4).
(4)$$p_{0}+\frac{\rho \nu^{2}}{2}=p_{L}+\frac{\rho \nu^{2}}{2}+\Delta p$$
where ν—flow velocity, $p_{0}$—initial point’s pressure (z=0), $p_{L}$—end point’s pressure (z=L), ρ—liquid density, $\Delta p=\lambda \frac{L}{d}\frac{\rho \nu^{2}}{2}$—pressure losses between initial and end points of a pipe, where λ—so-called the Darcy friction factor, L—length of the examined pipeline’ section, d—inner diameter of pipe.

In this operating state, the pressure distribution along the pipeline is described by a straight line, as shown in Figure 1 (green line). When considering the same pipeline with two leaks at points with coordinates $z_{u1}$ and $z_{u2}$, the pressure distribution is represented by three straight lines, as shown in Figure 1 (red solid line). It is worth noting the partial similarity of this distribution to the distribution corresponding to the occurrence of a single leak at a location with coordinate $z_{u}$ (red dotted line).

Using Bernoulli’s equation, a static model can also be obtained that includes two leaks. This model is composed of three equations: one for each part of the pipeline separated by leak positions, along with the locations of individual leaks on their periphery.

In general, static models are much simpler to design compared to dynamic models. They can easily be connected to measuring points on the pipeline. The solution of the static model requires significantly fewer computational resources to maintain the time regime between successive results of model calculations. In practice, limited computing resources can be a significant problem, particularly with dynamic flow models. Another important issue is the need to reduce computational overhead in simulations related to the flow process, as observed in [31].

### 2.2. Description of Proposed Diagnostic Methods

The proposed methods aim to diagnose double leaks using parameters derived from a mathematical flow model and parameters corresponding to the real pipeline. Based on differences between these parameters, their analysis leads to making decisions on detecting, locating, and estimating leakage size, including the issue of isolating the number of leaks that have occurred.

Both diagnostic methods use the following presets:-availability of pressure measurements $p_{1}$, …, $p_{N}$ along a pipeline, at N−1 pipeline segment ends, distinguished by the positions of N pressure sensors, and flow rate measurements $q_{1}$, $q_{2}$ at the pipeline’s inlet and outlet;-determination of parameters of both leaks: $z_{u1}$, $z_{u2}$—the coordinates of their positions, $q_{u1}$, $q_{u2}$—the magnitude (volumetric flows) of the leaks.

#### 2.2.1. Characteristics of Method I

The proposed solution makes use of a static liquid flow model to describe a pipeline with a leak event at two distinct locations. This approach assumes that measurements of the pressure and flow volume are served as the boundary conditions for solving the model, whereas the drop leak locations are used as parameters. Then, the pressure values $p_{mi}$ corresponding to pressure measurement points i=2, …, N−1 and the parameters of both leaks are established as the model outputs (see Figure 2).

The used pipeline model calculates pressure p(z) at any point along the pipeline according to the following expression:(5)$$p(z)=p_{0}-\lambda \frac{\rho \cdot q^{2}}{2\cdot A^{2}}\frac{z}{d}$$
where $p_{0}$—pressure at initial point (z=0), z—coordinate of pipeline’s length, d—inside diameter of a pipe, A—cross-sectional area of a pipe, q—volume flow rate, ρ—liquid density, λ—friction (linear loss) coefficient.

The model is based on the idea, which includes the determination of pressures for individual parts (segments) of the pipeline separated by the positions of the two considered leaks. Pressure at the inlet part of the pipeline, up to the position of the first leak $z_{u1}$, is defined using measurements *p1*, *q1*. Pressure at the outlet part of the pipeline, i.e., beginning from the position of the second leak $z_{u2}$ to the end of the pipeline, is defined relying on measurements $p_{N}$, $q_{2}$. As for the pressure between the positions of the leaks, it is determined according to a linear function stretched on $p_{u1}$, $p_{u2}$ pressures using the flow rate $q_{1}^{’}$ which is calculated based on $q_{1}$ and on the pressure gradient in this segment. Accordingly, the model uses the following equations:(6)$$p_{mi}=p_{1}-\lambda \frac{\rho \cdot q_{1}{}^{2}}{2\cdot A^{2}}\frac{\left(z_{i}-z_{1}\right)}{d}\;\mathrm{when}\;z_{i}<=z_{u1}$$
(7)$$p_{u1}=p_{1}-\lambda \frac{\rho \cdot q_{1}{}^{2}}{2\cdot A^{2}}\frac{\left(z_{u1}-z_{1}\right)}{d}$$
(8)$$p_{mi}=p_{N}+\lambda \frac{\rho \cdot q_{2}{}^{2}}{2\cdot A^{2}}\frac{\left(z_{N}-z_{i}\right)}{d}\;\mathrm{when}\;z_{i}>=z_{u2}$$
(9)$$p_{u2}=p_{N}+\lambda \frac{\rho \cdot q_{2}{}^{2}}{2\cdot A^{2}}\frac{\left(z_{N}-z_{u2}\right)}{d}$$
(10)$$q_{1}^{’}=\sqrt{\frac{2\cdot d\cdot A^{2}}{\lambda \cdot \rho }\cdot \frac{p_{u1}-p_{u2}}{z_{u2}-z_{u1}}}$$
(11)$$p_{mi}=p_{u1}-\lambda \frac{\rho \cdot q_{1}^{’}{{}_{}}^{2}}{2\cdot A^{2}}\frac{\left(z_{u2}-z_{u1}\right)}{d}\;\mathrm{when}\;z_{u1}<z_{i}<z_{u2}$$
(12)$$\begin{array}{l} q_{u1}=q_{1}-q_{1}^{’}\\ q_{u2}=q_{1}^{’}-q_{2} \end{array}$$

It is expected that the discussed model will make it possible to accomplish tasks aimed at determining the locations and magnitudes of the leaks. It can be effectively employed after a positive detection test which shows the presence of a leak event in a pipeline. In order to determine the leaks’ location and size, we propose to apply an objective function $F_{C}$ dependent on the hypothetical positions $z_{u1}$ and $z_{u2}$ of both leaks. The function is defined as the square deviation of the modeled pressures from the measured pressures (13). By identifying its minimum, the best model fitting to the measurements collected on the pipeline with real two leaks at the positions $z_{u1}$ and $z_{u2}$ (14) can be obtained.
(13)$$F_{C}=\sum_{i=2}^{N-1}{\left(p_{mi}-p_{i}\right)}^{2}$$
(14)$$\min F_{C}(z_{i(1)},z_{i(2)})\Rightarrow \left\{z_{u1},z_{u2}\right\}\;\mathrm{for}\;z_{1}<z_{u1}<z_{N},\;z_{u1}<z_{u2}<z_{N}$$

#### 2.2.2. Characteristics of Method II

The proposed solution also uses a static model of a pipeline. This approach is based on the assumption that the pressure drop along the segment of length L can be expressed as follows:(15)$$p_{0}-p_{L}=\lambda \frac{\rho \cdot q^{2}}{2\cdot A^{2}}\frac{L}{d}$$
where $p_{0}$—pressure at initial point (z=0), $p_{L}$—pressure at end point (z=L), L—length of pipeline’s segment, d—inside diameter of a pipe, A—cross-sectional area of a pipe, q—volume flow rate, ρ—liquid density, λ—friction (linear loss) coefficient.

Or based on pressure gradient gr:(16)$$\begin{array}{l} gr=\frac{p_{0}-p_{L}}{L}\\ gr=\lambda \frac{\rho \cdot q^{2}}{2\cdot A^{2}}\frac{1}{d} \end{array}$$

After the occurrence of a leakage at a point with $z_{u}$ coordinate situated in the *i*-th segment of the pipeline, the flow distribution takes the form presented in Figure 3. For comparison, Figure 4 presents such distribution corresponding to the identical pipeline’s operation but considered from the perspective of the consecutively measured pressures along the pipeline. Then, accordingly to relationship (15), the volume flow rate $q^{’}{}_{1}$ in the *i*-th segment of the pipeline takes an intermediate value between the pipeline’s inbound flow rate $q_{1}$ and outbound flow rate $q_{2}$.

In the course of transformations, relation (15) leads to the following equation enabling the calculation of the leak coordinate $z_{u}$:(17)$$z_{u}=z_{i}+(z_{i+1}-z_{i})\cdot \frac{q_{i}^{’}{{}_{}}^{2}-q_{i+1}{}^{2}}{q_{i-1}{}^{2}-q_{i+1}{}^{2}}$$

The coordinate $z_{u}$ of the leak point can be alternatively determined using pressure gradients as follows:(18)$$z_{u}=z_{i}+(z_{i+1}-z_{i})\cdot \frac{\lambda_{i-1}\cdot (\lambda_{i+1}\cdot gr_{i}-\lambda_{i}\cdot gr_{i+1})}{\lambda_{i}\cdot (\lambda_{i+1}\cdot gr_{i-1}-\lambda_{i-1}\cdot gr_{i+1})}$$
where assuming the notation for the pipeline’s segment as k, k∈<i−1,i+1>, then:(19)$$gr_{k}=\frac{p_{k}-p_{k+1}}{z_{k+1}-z_{k}}$$

While $\lambda_{k}$ is the friction (linear loss) coefficient for this segment.

Whereas the leakage intensity $q_{u}$ can be calculated according to the following relationship:(20)$$q_{u}=\sqrt{\frac{2\cdot d\cdot A^{2}}{\lambda_{i-1}\cdot \rho }\cdot gr_{i-1}}-\sqrt{\frac{2\cdot d\cdot A^{2}}{\lambda_{i+1}\cdot \rho }\cdot gr_{i+1}}$$

To determine parameters $z_{u}$, $q_{u}$ of a leak using the presented method, a crucial question is to understand that pressure at two pipeline points before the leak spot as well as at two points behind the leak spot must be available. Moreover, the pipeline’s parameters must be also known.

A direct application of the discussed method for double leak events is possible when both leaks do not occur in the same pipeline segment or in two adjacent segments. For this purpose, we can consider two separated (not adjacent) pipeline segments with a single leak, applying Equations (18)–(20) to each of them. In the case when the leaks occur in adjacent segments, further equations are necessary.

The key to performing diagnostics designed in such a way is the ability to establish that there are two actual leaks in the pipeline.

In order to facilitate such an inference, we propose index functions $IG_{i-j}$ which are defined as the difference between the pressure gradient in the segment i−1 (that corresponds to pressure sensors denoted i−1 and i) and the segment j (that corresponds to pressure sensors denoted j and j+1), where i∈<2,N−1>, j∈<i,N−1>.

For the leak detection, these individual index functions $IG_{i-j}$ are compared with their corresponding threshold values, which are determined for the pipeline operating without a leak. When a given function $IG_{i-j}$ exceeds its assumed threshold value, it can indicate the occurrence of a leakage between i and j sensors.

In the case of a single leak, the indicators considered for adjacent pipeline segments across sensor i, and therefore marked as $IG_{i-i}$, exceed threshold values only for two consecutive indices i. For such a possibly identified area, the segment with a leak is, in fact, either the initial or final segment. In the case of two leaks occurring in locations assigned to two different segments, these indicators achieve cross-border values for between two and four indices i. Examining the values of the individual indicators should allow recognizing the segments in which leaks occurred. In the case when the considered segments are not adjacent, we can decompose the pipeline into two single-leak segments so as to enable easier direct application of the proposed method.

#### 2.2.3. Minimum Requirements for Implementation of Methods

Implementation of both proposed methods requires accessing additional pressure measurement points along the pipeline, in addition to measurements usually collected at its inlet and outlet. Their minimum number is determined by the need to identify the actual distribution of the pressure (Figure 1) in the sense of distinguishing between the occurrence of a single (dotted red line) or double (solid red line) leak.

Both methods require at least four additional pressure measurement points. The first point should be in front of the first leak, the second and third between both leaks and the fourth after the second leak. In the case of the second method, based on the pressure gradient calculation, the first one, together with the measurement point at the pipeline’s inlet, will allow calculating the initial pressure gradient. The next two measurement points, located between both leaks, will allow to calculate the intermediate pressure gradient. The fourth of these points, located downstream of the second leak, together with the measuring point at the pipeline’s outlet, will allow the final pressure gradient calculation.

Since leaks can occur anywhere in the pipeline, therefore, to meet the above requirements, many more measuring points are required.

### 2.3. Measurement Data from Model Pipeline

The methods presented above were verified using measurement data gathered during tests performed on a laboratory pipeline that pumped water.

#### 2.3.1. Stand with Model Pipeline

The laboratory pipeline is nearly 400 m in length, including the main pipe segment (Figure 5), which is 380 m long, with appointed inlet coordinate z=0 and outlet coordinate z=380. This section is built of polyethylene tubes (HDPE) with an internal diameter of 34 mm and an external diameter of 40 mm.

On the pipeline, typical measuring devices have been installed, including two electromagnetic volume flow sensors (at the inlet and outlet), nine pressure sensors (at several points along the pipe), as well as two temperature sensors. The measuring devices send signals to a PC equipped with a 16-bit A/D converter (see Table 1).

#### 2.3.2. Experiments with Simulated Double Leakages

On the pipeline, the experiments with simulated two leaks were performed. For this purpose, electromagnetic valves mounted at points with coordinates $z_{leak}(L1)=155$ m and $z_{leak}(L2)=315$ m were used (see Figure 6). Two different cases of double leaks, depending on their occurrence time scheme, were considered: simultaneous and non-simultaneous. The former ones were simulated at the same time during pipeline’s operation in steady state conditions when water was pumped with an intensity of about $q_{nom}\approx 140$ L/min. By contrast, the latter ones were simulated, keeping identical conditions for the first leak, but the second leak was initiated at the moment when the pressure wave that arose as a result of the first leak reached its position (see Table 2). Each of the leaks was simulated by suddenly opening the relevant valve. The intensities of both leaks amounted to about 0.85–1.49 L/min. They were measured using a cylinder and a stopwatch. During the experiments, the signals generated by the above-mentioned pressure and flow sensors were sampled with a frequency of $f_{P}=100$ Hz. Each experiment with the same simulated leakage configuration was repeated twice.

## 3. Results and Discussion

Using the known geometrical and operational parameters of the laboratory pipeline, static flow models and diagnostic algorithms provided for both proposed methods were implemented in the MATLAB environment. Their verification was carried out by running the MATLAB code using the measurement data collected during each of the experiments with simulated types of two leaks.

### 3.1. Verification of Method I

The verification of Method I involved the initial testing phase, i.e., calibration. In the next step, the possibilities in terms of detection, localization, and estimation of the intensity of simulated double leaks were investigated.

#### 3.1.1. Initial Testing Phase

In this phase, certain initial measurements $p_{i}(i\in <1,\;N>)$, $q_{1}$, $q_{2}$ related to the pipeline’s operation under steady state conditions without a leak were used. Based on these measurement data, the corresponding $ps_{i}$, $qs_{1}$, $qs_{2}$ variables were obtained using time windows covering 100 signal samples. Subsequently, leak detection procedures and leak identification procedures were calibrated by determining individual variables’ statistics and leak detection thresholds. The loss coefficient λ for all segments of the pipeline was also determined relying on dependence (5).

#### 3.1.2. Leak Detection

In order to detect an emergency state (a leak), the approach typically used for single-leak diagnosis was adopted. The detection algorithm based on the PPA method (pressure point analysis) was used, which involved calculation of index functions $IF_{i}$ derived from comparing in time $ps_{i}$ variables, where i∈<2, 8>.

In the case when particular index functions $IF_{i}$ exceed their corresponding alarm thresholds, they indicate pressure anomalies in a pipe. Such crossing moments in relation to the moment of the first leak occurrence are defined as leak detection times $T_{w}$. Table 3 presents these time values obtained through $IF_{3}$ to $IF_{7}$ index functions for all performed experiments.

While analyzing the resultant detection times, it can be stated that the most useful were indicators $IF_{4}$, $IF_{6}$ corresponding to $p_{4}$ or $p_{6}$ pressure measurement points because they informed the most quickly about the leak. These pressure measurement points were positioned in direct proximity to the spots of both leaks.

We can also conclude that the detection times observed in individual experiments indicate the presence of a transient state of a pumping process. On a theoretical basis, such a state includes the propagation of pressure waves in the pipeline induced by each of the leak events. However, the determined values of index functions $IF_{i}$ corresponding to different sensors did not provide grounds to indicate the occurrence of two separate leaks. This problem concerned both types of double leaks simulated during the experiments, i.e., concurrent (samples: 1, 2, 5, 6) and non-concurrent (samples: 3, 4, 7, 8) leaks.

On the other hand, we can notice a certain regularity that consists in a shortening of detection times $T_{w}$ for leakages of enlarged magnitudes. Additionally, detection times $T_{w}$ were getting longer in the case of non-concurrent leaks, as compared to the ones obtained in the experiments with simultaneous leaks. This could be explained by a delayed in time effect of the accumulation of pressure waves caused by both leaks.

#### 3.1.3. Determination of the Location and Magnitude of Leaks

Leak localization procedure was activated 4.5 s after confirming the presence of a leak in a pipeline, i.e., already in new steady-state conditions. It was carried out in a 50-s timeframe by sequentially initiated computations of the mathematical pipeline model with assumed two leaks and using the algorithm aiming at minimization of the objective function (13). The calculations employed the measurement data provided as input at intervals of one second.

Figure 7 presents exemplary courses of $z_{u1}$ and $z_{u2}$ leak location and leakage intensity $q_{u1}$ and $q_{u2}$ in the form of averaging five subsequent cycles of calculation obtained for experiment 1 (see Table 2).

The obtained results of leak localization and leak size estimation, expressed as average values for the entire above-mentioned 50-s period, are presented in Table 4. It should be mentioned here that the values representing the estimated size of both leakages are given with a resolution of 0.1 L/min associated with the metrological characteristics of the measuring devices (i.e., electromagnetic volume flow meters) used on the examined pipeline.

Analyzing the results obtained for individual experiments, we can ascertain that errors expressed by relationship (21) in determining the locations $z_{u1}$, $z_{u2}$ of both leaks vary between −9.8 and −20.4 m, and, respectively, between 7.7 and 15.7 m. These error intervals can be recognized as satisfactory. As regards the errors in estimating the leak sizes $q_{u1}$ and $q_{u2}$, they amount from −0.1 to 0.1 L/min, as well, respectively, from 0.1 to 0.4 L/min, which corresponds to a similar or even higher assessment mark.
(21)$$ERx=x_{u}-x_{leak}$$

### 3.2. Verification of Method II

Verification of Method II, carried out identically as for Method I, included in the case of each of the experiments an initial testing phase, i.e., calibration, and later investigation of the possibilities of detection, localization, and estimation of the intensity of simulated double leaks.

#### 3.2.1. Initial Testing Phase

In general, this stage was identical as in the case of Method I. The loss coefficient $\lambda_{k}$ was also determined for individual segments of the pipeline using relation (5).

#### 3.2.2. Leak Detection

In the initial phase, we adopted the same approach as used for Method I. What is more, index functions $IG_{i-i}$ (where i∈<2,8>), which define gradient differences for adjoining segments, were scrutinized. The results are presented in Table 5 in the form of a single row (sample number not being provided), which is sufficient due to the similarities observed in all experiments. Regarding Table 5, it should be explained that the row labeled ‘segment’ contains the numbering of segments being tested by means of the particular index function. Furthermore, the ‘+’ and ‘–’ signs in the row labeled ‘result’ mean in the case of an individual index function, respectively, exceeding or not exceeding the corresponding threshold value. The obtained results demonstrate the occurrence of a leak in segments indexed k=4 and k=6, which are not situated in a direct neighborhood. Taking the above into account, in the next diagnostic step, these selected segments were used for the exact leak localization procedure.

In order to confirm such findings, index functions $IG_{i-j}$ were additionally analyzed, where i∈<2, 7> and j=i+1. These indicators express differences in pressure gradients corresponding to pipeline segments, taking into consideration both ends of any given segment k=i. As a consequence, segments with leaks were identified. In practice, this identification came after the detection of a leak event in the pipeline. Table 6 presents times that correspond to the resolution of such an identification obtained for the considered individual index functions.

These times, in the case of simultaneously simulated leaks the valued ranged from 0.8 to 1.1 s, whereas in the case of leaks simulated at different moments the values ranged from 1.0 to 2.2 s, counting from the moment of the occurrence of the first leak event.

#### 3.2.3. Location and Magnitude of Leaks

Leak localization was performed in two modes corresponding to different operational conditions of a pipeline, i.e., a steady state and a transient state.

*Mode 1—steady state.* Leak localization procedure was activated in the same time regime as for Method I, i.e., after 4.5 s from the discovery of a leak event, which corresponded to a new steady state condition in the pipeline. This was carried out in a 50-s timeframe by sequentially initiating computations according to Equations (18)–(20), taking into account previously identified leaks for the pipeline segments k=4 and k=6. The calculations were performed cyclically every 1-s interval, with the collected measurement data being provided to the localization procedures at each cycle. Figure 8 shows the course of leak parameters: locations $z_{u1}$, $z_{u2}$, and intensities $q_{u1}$ and $q_{u2}$, in the form of an average over five consecutive calculation cycles for sample 1. The values of the leak parameters obtained for individual experiments, expressed as average values over the above-mentioned whole 50-s time period, are presented in Table 7. In addition, the penultimate and last column of Table 7 show, respectively, total loss (22) calculated as the difference between the flow rates measured at the inlet and outlet of a pipeline, and error (23), which reflects how much the determined intensities of both leaks considered as their sum differ with respect to the determined balance between the flow intensities measured at both ends of the pipeline.
(22)$$qu=q_{1}-q_{2}$$
(23)$$\Delta qu=qu-(q_{u}{}_{1}+q_{u}{}_{2})$$

While analyzing the results obtained for individual experiments, we can notice that the errors in determining both leak locations $z_{u1}$, $z_{u2}$ expressed according to relationship (21) ranged between −2.0 and −6.3 m, and, respectively, between 3.4 and 8.1 m. Whereas the errors in determining both leak sizes $q_{u1}$, $q_{u2}$ varied from −0.1 to −0.2 L/min and from 0.0 to 0.2 L/min, respectively. The errors (differences) between the sum of the determined leak flows and the balanced liquid flows were between 0.1 and 0.3 L/min. The resulting values of estimated leak parameters (location, intensity) can be considered even better than the ones obtained with Method I.

*Mode 2—transient state.* Here, the leak localization procedure was started immediately after the leaks were isolated with $IG_{4-5}$, $IG_{6-7}$ functions as occurring in the pipeline segments k=4 and k=6. In the case of individual experiments, the computations were initiated at the moment which corresponded to the greater of the two times given in Table 6. The calculations were performed cyclically every 50 ms in 8-s timeframes, with the collected measurement data being provided to the procedures at each cycle. Figure 9 presents the course of the resulting leak locations $z_{u1}$, $z_{u2}$ and leak intensities $q_{u1}$, $q_{u2}$ in the form averaged over 10 consecutive calculation cycles for experiment 1 (see Table 2).

Based on result values of leak location and intensity estimation pertaining to both leakages, we can observe that they yield useful diagnosis only after a certain period of time counting from the above-mentioned starting point of the computations. In the case of leak localization, the useless time intervals ranged from about 1.2 to 1.8 s for the first leak at 155 m and, respectively, from about 1.8 to 2.1 s for the second leak at 315 m.

The visible variability of the values obtained results from established forms of calculation Formulas (18) and (20) that use pressure gradients calculated from pressure measurements. In the initial phase, the impact of large pressure changes is visible, which are associated with the transients caused by the occurrence of both leaks and the overlap of these phenomena. Once transients disappear and stable flow conditions are obtained again, the resulting values are already more unambiguous. However, they are not fully stabilized due to the use of time windows with a smaller amount of data. On the one hand, this allows for faster results over time in relation to the initial moment included in such a window. On the other hand, there is an effect of less accurate averaging than in the case of time windows covering a longer time interval and, at the same time, a larger amount of data.

Additionally, it was concluded that for samples corresponding to non-concurrent leaks (experiments: 3, 4, 7, 8), the localization estimate of the first leak $z_{u1}$ became more stable after longer time in comparison to samples corresponding to concurrent leaks (experiments: 1, 2, 5, 6). On the other hand, the unstable periods of the localization estimate of the second leak $z_{u2}$ for both types of simulated leaks did not differ significantly. This is caused by the fact that a pressure wave, which in the case of simulated non-concurrent two leak events is provoked by the second of them, exerts a time-shifted impact on the conditions of the pumping process.

### 3.3. Possibility of Results Improvement

The authors assume that the efficiency of both proposed methods for diagnosing double leaks can be further developed by improving static flow models. For this purpose, the solutions presented in paper [21] may be helpful, as they propose a more accurate description of flow than the Darcy–Weisbach equation approach. In practice, instead of using previously accepted constant values for friction coefficients for individual parts of the pipeline separated by the two leaks under consideration, these coefficients would become more realistic. This can be achieved by using a number of calculation formulas proposed in the literature.

Moreover, the authors recognize the need to supplement both methods with more advanced algorithms aimed at identifying not just single leaks but double leaks in the pipeline. For this purpose, algorithms [32] previously developed by one of the authors of this work may be useful. These algorithms are designed to detect the phenomenon of pressure wave propagation and identify their fronts and have been tested on single leaks. Since the occurrence of double leaks, particularly when they are small in size, and their transients overlap, can significantly blur the readability of such a phenomenon, alternative solutions for identifying leaks may also be considered.

Another matter that the authors consider as a direction for further works is to determine the effect of the number and location of additional intermediate pressure measurement points on the accuracy of both methods.

## 4. Conclusions

The article presents two methods of diagnosing double leaks which were experimentally verified using measurement data collected from the laboratory water pipeline. The methods, along with dedicated algorithms, proved useful for diagnosing small leaks with intensities ranging from 0.6 to 1% of the nominal flow rate, simulated at two different points of the duct.

The proposed methods, due to low computational complexity, good efficiency, and reliability, can be considered in the case of specific pipeline operating conditions and double leakage characteristics as an alternative to model-based methods that use dynamic flow models. Experimental results also suggest that both methods should be able to handle double leakage events in real pipelines, even during other typical operating conditions changes.

In further work, a more extensive examination of the effectiveness of the discussed methods is intended. This involves addressing such questions as even smaller intensity of leaks and their variable (e.g., increasing) character in time, allowing for a more realistic representation of cases when pipe damage is progressing.

## Figures and Tables

**Figure 1 sensors-23-07751-f001:**
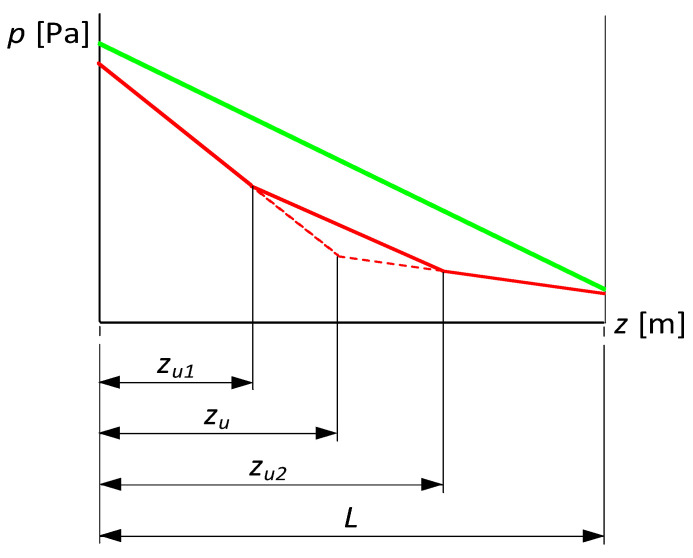
Pressure distribution in no-leak conditions (green line) and after single (red dotted line) and double leak occurrence (red solid line).

**Figure 2 sensors-23-07751-f002:**
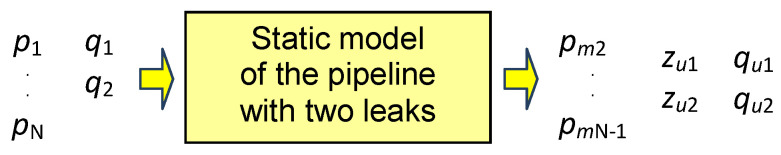
Model’s data flow diagram.

**Figure 3 sensors-23-07751-f003:**
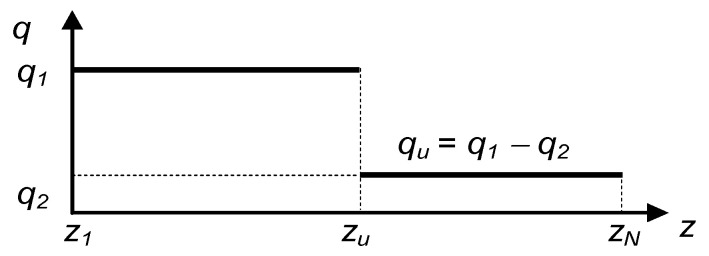
Distribution of volume flow rate *q* in the pipeline with a leak of coordinate *z_u_* corresponding to stable conditions.

**Figure 4 sensors-23-07751-f004:**
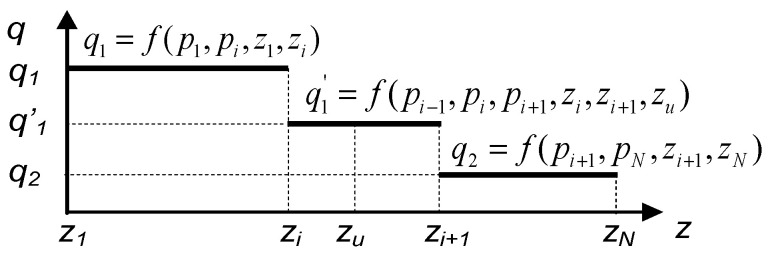
Distribution of volume flow rate *q* based on pressure measurements in the pipeline with a leak of coordinate *z_u_* in the *i*-th sensor’s segment.

**Figure 5 sensors-23-07751-f005:**
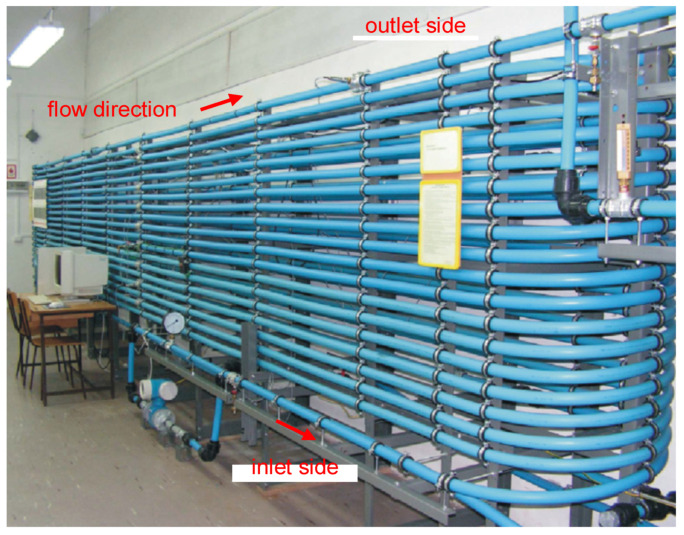
Laboratory pipeline model.

**Figure 6 sensors-23-07751-f006:**
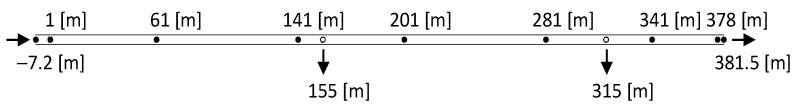
Pressure transmitters and leak point positions.

**Figure 7 sensors-23-07751-f007:**
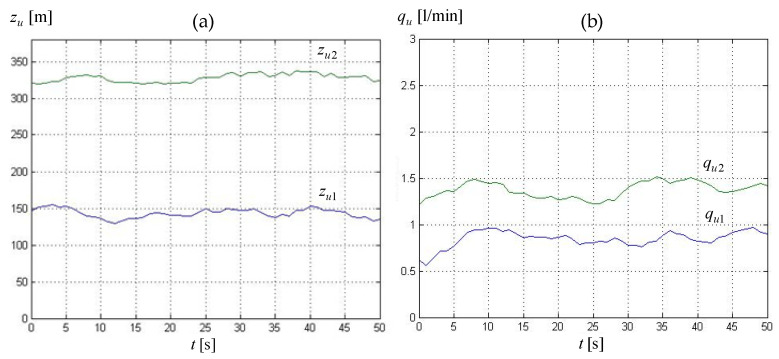
Leakage parameters evaluated in consecutive diagnostic procedure cycles obtained by Method I and data for experiment 1 (‘0’ on the time axis corresponds to the 180-th second of experiment data recording): (**a**) leak locations *z_u_*_1_, *z_u_*_2_, (**b**) leak intensity *q_u_*_1_, *q_u_*_2_.

**Figure 8 sensors-23-07751-f008:**
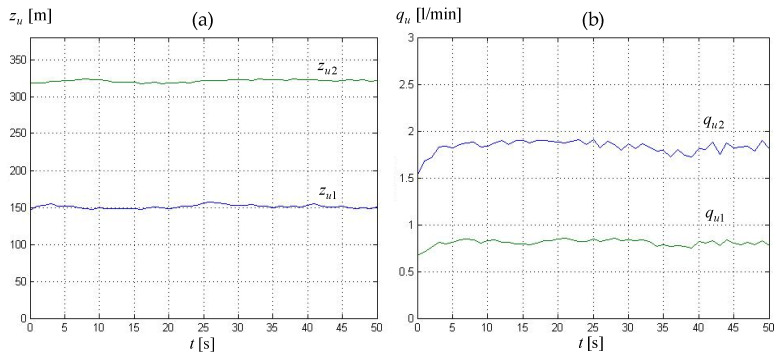
Leakage parameters evaluated in consecutive diagnostic procedure cycles obtained by Method II and data for experiment 1 (‘0’ on the time axis corresponds to the 180-th second of experiment data recording): (**a**) leak locations *z_u_*_1_, *z_u_*_2_, (**b**) leak intensity *q_u_*_1_, *q_u_*_2_.

**Figure 9 sensors-23-07751-f009:**
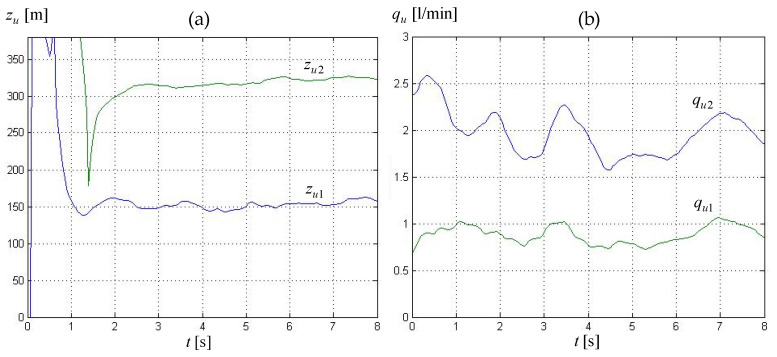
Leak parameters in transient state obtained using Method II and data for experiment 1 (‘0’ on the time axis corresponds to the 180-th second of experiment data recording): (**a**) leak locations *z_u_*_1_, *z_u_*_2_, (**b**) leak intensity *q_u_*_1_, *q_u_*_2_.

**Table 1 sensors-23-07751-t001:** Characteristics of measurement system.

Measuring Devices	Pressure Sensors			Flow Rate Sensors
Location [m]	$p_{1}=-7.2$	$p_{2}=1$	$p_{3}=61$	$q_{1(in)}=-6.5$ $q_{2(out)}=382.2$
	$p_{4}=141$	$p_{5}=201$	$p_{6}=281$	
	$p_{7}=341$	$p_{8}=378$	$p_{9}=381.5$	
Range	0–10 [bar]			0–200 [L/min]
Accuracy	0.1% of range			0.2% of range
Summary Uncertainty(sensor + A/D converter)	±0.012 [bar]			±0.44 [L/min]

**Table 2 sensors-23-07751-t002:** Simulated leakage parameters.

No.	Simulated Leaks		$\mathrm{Coordinate}\;z_{leak}$ [m]	Occurrence Time [s]	$q_{leak}$	
					[L/min]	$[\%]\;q_{nom}$
1	Concurrent	L1	155	180.00	0.96	0.69
2		L2	315	180.00	1.08	0.77
3	Non-concurrent	L1	155	180.00	0.85	0.71
4		L2	315	180.50	1.07	0.71
5	Concurrent	L1	155	180.00	1.49	1.06
6		L2	315	180.00	1.21	0.86
7	Non-concurrent	L1	155	180.00	1.49	1.06
8		L2	315	180.50	1.19	0.85

**Table 3 sensors-23-07751-t003:** Method I—response times obtained for used detection functions (the shortest time for each experiment in bold).

No.	$\mathrm{Response}\;\mathrm{Time}\;T_{w}$ [s]				
	*IF* _3_	*IF* _4_	*IF* _5_	*IF* _6_	*IF* _7_
1	1.25	0.72	0.72	**0.66**	0.69
2	0.82	**0.57**	0.61	0.58	0.60
3	1.28	**0.90**	0.95	0.99	1.02
4	1.51	**0.88**	0.95	0.89	0.96
5	1.10	0.60	0.60	**0.54**	0.55
6	0.81	**0.34**	0.49	0.55	0.63
7	0.95	**0.59**	0.67	0.76	0.83
8	0.84	**0.47**	0.60	0.76	0.86

**Table 4 sensors-23-07751-t004:** Method I—results of the leak localization and leak size estimation.

No.	Leak Localization [m]		Leak Intensity [L/min]	
	$z_{u1}$	$z_{u2}$	$q_{u1}$	$q_{u2}$
1	142.9	327.6	0.9	1.4
2	142.8	326.3	1.0	1.2
3	134.6	323.8	0.9	1.2
4	139.7	322.7	0.8	1.2
5	145.2	327.3	1.4	1.5
6	140.9	326.0	1.4	1.4
7	145.1	330.7	1.4	1.6
8	137.4	324.5	1.5	1.4

**Table 5 sensors-23-07751-t005:** Method II—leak detection results with *IGi–i* index functions corresponding to adjoining pipeline segments.

Indicator	*IG_2–2_*	*IG_3–3_*	*IG_4–4_*	*IG_5–5_*	*IG_6–6_*	*IG_7–7_*	*IG_8–8_*
Segment	1–2	2–3	3–4	4–5	5–6	6–7	7–8
Result	–	–	+	–	+	+	–

**Table 6 sensors-23-07751-t006:** Method II—leak detection time obtained from the index functions of segments k=4 ($IG_{4-5}$) and k=6 ($IG_{6-7}$).

No.	Response Time *T_w_* [s]	
	*IG_4–5_*	*IG_6–7_*
1	1.1	0.7
2	1.1	0.8
3	1.8	1.1
4	2.2	1.1
5	1.1	0.6
6	0.8	0.7
7	0.6	1.0
8	0.6	1.0

**Table 7 sensors-23-07751-t007:** Method II—results of the leak localization and leak intensity estimation.

No.	Leak Localization [m]		Leak Intensity [L/min]			
	$z_{u1}$	$z_{u2}$	$q_{u1}$	$q_{u2}$	qu	Δqu
1	151.1	321.3	0.8	1.2	2.2	0.2
2	153.0	321.4	0.9	1.2	2.2	0.1
3	149.9	320.0	0.8	1.1	2.1	0.2
4	148.7	318.4	0.7	1.1	1.9	0.1
5	148.7	318.4	1.3	1.4	2.9	0.2
6	151.3	320.9	1.3	1.3	2.8	0.2
7	151.8	323.1	1.3	1.4	3.0	0.3
8	150.7	321.1	1.3	1.4	2.8	0.1

## Data Availability

Not applicable.

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
