# Peer review of "Two-Leak Case Diagnosis Based on Static Flow Model for Liquid Transmission Pipelines"

_sensors, 2023, doi:10.3390/s23187751_

Round 1
Reviewer 1 Report
The manuscript is interesting and quite well-written. If the authors consider my remark, I will suggest the manuscript should be accepted for publication in the Journal Sensors.
The authors propose methods that presuppose the availability of pressure measurements along the pipeline. In practice, while it is common for inlet and outlet pressures to be measured, pressure measurements are less common to be available at intermediate points in the pipeline. Please, analyze this in-depth, clarifying aspects such as: how many minimum measurement points would be necessary and how the number and location of these points could affect the precision of the proposed methods.
Author Response
We kindly send our answers in the attached file.

Reviewer 2 Report
This article proposes two dual leakage diagnosis methods suitable for liquid transportation pipelines. Both methods are suitable for detecting both simultaneous and non-simultaneous leaks. Both methods use static type pipeline flow models. The first method defining the square deviation between the modeling pressure and the measured pressure on the pipeline as the objective function and minimizing it. The second method combines gradient indicators and algorithms for leak detection for leak diagnosis. Overall, this article provides a comprehensive description and has certain theoretical guidance significance for the leakage of liquid pipelines. However, there are still some issues that need to be modified and improved in this article. The detailed comments are as follows:
1. The description in the fifth paragraph of the introduction is incorrect. Specifically, for these two types of multiple leaks, this method may result in poor positioning results.
2. In section 2.1.4, it is possible to consider changing curves "0" and "1" to curves " " and "
" respectively, in order to better correspond to Fig. 1.
3. The range of pressure sensors in Table 1 should be 0-10, and the range of flow sensors should be 0-200.
4. The description in section 2.3.2 does not match Table 2. According to the experimental data in Table 2, the intensity of the two leaks is 0.85-1.49 l/min. In addition, the symbol (÷) in section 2.3.2 is used incorrectly.
5. The results of 6-7 in Table 5 are "+", indicating that a leak has occurred in this segment. Why not consider this segment? Please explain.
Author Response
We kindly send our answers in the attached file

Reviewer 3 Report
Paweł Ostapkowicz and Andrzej Bratek's article "Diagnosis of two leaks based on a static flow model in liquid pipelines" deals with the problem of diagnosing multiple leaks in liquid pipelines using analytical methods. The authors emphasize that traditional approaches based on mathematical models can be complex and time-consuming. To solve this problem, they propose two simplified methods for detecting double leaks, applicable to both simultaneous and non-simultaneous leaks. The effectiveness of both methods has been demonstrated in tests using experimental plumbing data.
This article presents an intriguing and practical solution to the complex problem of diagnosing multiple leaks in fluid pipelines. The authors recognize the complexity of mathematical models and advocate the use of simpler but more effective approaches. The proposed methods are described in a clear and understandable way, which makes them accessible to readers with different levels of experience in this field.
The strength of the study is that it draws on experimental data on plumbing, as it demonstrates the practicality and effectiveness of the proposed methods.
The general organization of the article and its design are satisfactory.
In conclusion, the work "Diagnostics of two leaks based on a static flow model in pipelines for the transport of liquids" makes a valuable contribution to the field of leak detection in pipelines. The simplicity and high efficiency of the proposed methods, demonstrated on experimental data, make this article of interest to researchers and practitioners in the field of pipeline integrity and maintenance.
Author Response
We kindly thank you for your positive reviewReviewer 4 Report
There are too many hints and self-praise about the good methods in the abstract. However, the methods themselves are not described. However, the abstract should reflect the content of the paper in an abbreviated form.
Line 34: very incomplete list, please also include elaborated methods like gas detection sensors, TVOC, infrared analysis, cameras, acoustic monitoring, etc.
Line 53-71: is actually without content and contains many repetitions and is very little concrete.
71-102: very unspecific statements, what does the reader want to do with it, ? Maybe delete the paragraph
104 to 112 also no statement except that measurements improve the situation...,
112- 159: very many repetitions, shorten massively.
Overall, the models do not become clear. Nor does it become clear where meaningful accuracies are in real environmental conditions.
The theory initially includes a lot of general knowledge described in textbooks.
The article is not able to elaborate the particular merits of the touted method.
The authors should retract the article, improve thoroughly and resubmit it.
Author Response
Dear Reviewer,
We have corrected the abstract.
We have added information on different category of solutions, so-named as external, direct, or hardware based methods, in the second paragraph of the introduction.
We have corrected the introduction in terms of eliminating repetitions.
In addition, we supplemented the text (new subchapter 2.2.3) with an indication of the requirements regarding the minimum number of measuring sensors necessary to obtain a correct diagnosis. Besides, in the supplementary subchapter 3.3, we have indicated significant problems and challenges and outlined directions for future work that may enable further improvement of efficiency.
Regards,
Authors
Round 2
Reviewer 4 Report
should be published in present form